Resource

# Tradeoff between metabolic i-proteasome addiction and immune evasion in triple-negative breast cancer

Alaknanda Adwal[1] , Priyakshi Kalita-de Croft[2] , Reshma Shakya[3], Malcolm Lim[2], Emarene Kalaw[2], Lucinda D Taege[2,4], Amy E McCart Reed[2] , Sunil R Lakhani[2,4] , David F Callen[5,*], Jodi M Saunus[2,*]

In vitro studies have suggested proteasome inhibitors could be effective in triple-negative breast cancer (TNBC). We found that bortezomib and carfilzomib induce proteotoxic stress and apoptosis via the unfolded protein response (UPR) in TNBC cell lines, with sensitivity correlated with expression of immuno-(*PSMB8/9/10*) but not constitutive-(*PSMB5/6/7*) proteasome subunits. Equally, the transcriptomes of i-proteasome–high human TNBCs are enriched with UPR gene sets, and the genomic copy number landscape reflects positive selection pressure favoring i-proteasome activity, but in the setting of adjuvant treatment, this is actually associated with favorable prognosis. Tumor expression of *PSMB8* protein (β5i) is associated with levels of MHC-I, interferon-γ–inducible proteasome activator PA28β, and the densities of stromal antigen-presenting cells and lymphocytes (TILs). Crucially, TILs were protective among TNBCs that maintain high β5i but did not stratify survival amongst β5i-low TNBCs. Moreover, β5i expression was lower in brain metastases than in patient-matched primary breast tumors (n = 34; *P* = 0.007), suggesting that suppression contributes to immune evasion and metastatic progression. Hence, inhibiting proteasome activity could be counterproductive in the adjuvant treatment setting because it potentiates anti-TNBC immunity.

## Introduction

The ubiquitin-proteasome system (UPS) is vital for cellular homeostasis, acting to regulate protein expression and eradicate oxidized, misfolded proteins. The proteasome is its 2.5-MD catalytic engine—the major cellular protein recycling complex that degrades poly-ubiquitinated substrates (Sorokin et al, 2009). Its 20S core comprises 28 subunits arranged in four heptameric rings—two outer "alpha" and two inner "beta" rings surround a central pore through which unfolded proteins pass as they are cleaved. Three

subunits (six per core) provide a complete spectrum of proteolytic activity after acidic, basic, and hydrophobic amino acids: β1 caspase-, β2 trypsin-, and β5 chymotrypsin-like (encoded by *PSMB6*, *PSMB7*, and *PSMB5*). Proteasome substrate specificity, throughput, and subcellular localization are regulated by activator complexes that dock with the core, forming single- and double-capped holoenzymes. A 19S cap is essential for de-ubiquitination, translocation, and ATP-dependent substrate unfolding. Under normal conditions, 19S-capped 20S core complexes the predominant proteasome species present in the cell ("constitutive" proteasomes). However, in certain conditions, the core can also dock with PA28α/β, PA28γ, and PA200 caps, which lack catalytic activity but widen the 20S pore and increase throughput (reviewed in Kors et al [2019]).

In addition to posttranslational regulation and buffering against proteotoxic stress, the UPS is also the major source of peptides presented to the immune system via MHC class-I molecules. Accordingly, IFN-γ is a potent stimulus for proteasome compositional change. IFN-γ down-regulates constitutive proteasome subunit genes and induces the genes that encode PA28α and PA28β (*PSME1* and *PSME2*) and alternative 20S core subunits β1i, β2i, and β5i (*PSMB9*, *PSMB10*, and *PSMB8*), which increases proteasome throughput and produces a varied repertoire of MHC-I epitopes via different cleavage site preferences (Aki et al, 1994). Hence, in the context of an immunologic challenge, IFN-γ–stimulated cells express a variety of hybrid "immuno(i)-proteasome" complexes.

Proteasome inhibitors (PIs) such as bortezomib and carfilzomib are effective for treatment of multiple myeloma and mantle cell lymphoma. Their clinical activity is partly attributed to disruption of the UPS, which stabilizes apoptotic proteins (e.g., p53 and Bcl2) and IκBα, inhibitor of the NFκB pro-survival pathway. Both inhibitors bind to β5 and β5i subunits, but carfilzomib binds irreversibly and has a favorable side effect profile (Cromm & Crews, 2017). PIs also exploit metabolic addiction to the proteasome. By promoting accumulation of misfolded and oxidized protein aggregates in the ER, they trigger the unfolded protein response (UPR), a sensing mechanism

[1]The Robinson Research Institute, Adelaide Medical School, The University of Adelaide, Adelaide, Australia   [2]The University of Queensland (UQ) Centre for Clinical Research, Faculty of Medicine, The University of Queensland, Brisbane, Australia   [3]QIMR Centre for Immunotherapy and Vaccine Development, Tumour Immunology Laboratory, QIMR Berghofer Medical Research Institute, Brisbane, Australia   [4]Pathology Queensland, The Royal Brisbane and Women's Hospital, Brisbane, Australia   [5]School of Medicine, Faculty of Health Sciences, The University of Adelaide, Adelaide, Australia

Correspondence: alaknanda.emery@adelaide.edu.au; j.saunus@uq.edu.au
*David F Callen and Jodi M Saunus contributed equally to this work

that promotes apoptosis in the event of unsustainably high rates of cellular metabolism (Sorokin et al, 2009). In cancer, rapid division of cells with mutated genomes *should* incite the UPR, but strategies evolve to cope with proteotoxicity, including proteasome over-expression (Adams, 2004). The UPR-inducing activity of PIs is attributed mainly to inhibition of constitutive proteasomes, based on a persisting assumption that this complex is the most abundant in non-hematologic tissues (Roeten et al, 2018). However, solid tumors also express the i-proteasome (Altun et al, 2005; Ho et al, 2007), and this could represent an efficient means for maintaining a concentration of toxic protein aggregates below the UPR induction threshold (Shringarpure et al, 2003).

Triple-negative breast cancers (TNBCs) comprise 15–20% of invasive breast carcinomas. They lack clinically significant expression of first-line systemic drug therapy targets (estrogen receptor, human epidermal growth factor receptor-2 [HER2]), are typically high-grade, are metabolically active, and often exhibit basal-like and/or mesenchymal phenotypes (Perou et al, 2000; Prat et al, 2010). Cytotoxic chemotherapy is a mainstay of clinical management, but 40–80% of patients still experience distant relapse and premature death, often involving visceral and brain metastases (Fulford et al, 2007; Pusztai et al, 2019). TNBC exhibits considerable molecular and clinical heterogeneity, unsurprising given theirs is a diagnosis of exclusion. Identifying biomarkers and therapeutic targets remain top research priorities. Promising recent developments have centered on exploiting the predictive and therapeutic significance of tumor-infiltrating lymphocytes (TILs). There is now international consensus that the presence and density of TILs in pretreatment diagnostic specimens predicts the response to neoadjuvant chemotherapy and also to second-line treatment where a low-moderate residual disease burden remains (Loi et al, 2019; Luen et al, 2019). Thus, despite the systemic immunosuppression experienced by patients undergoing active treatment, the presence of TILs before this indicates the capacity of the host immune system to synergize with chemotherapy.

In vitro and in vivo studies suggest that the proteasome is a potential therapeutic target in TNBC (Cardoso et al, 2004; Petrocca et al, 2013; Weyburne et al, 2017); however, preliminary clinical data from metastatic patients have been mixed (Yang et al, 2006; Engel et al, 2007; Schmid et al, 2008), and these trials pre-dated our understanding of the role of antitumor immunity in TNBC. This study was undertaken to establish the molecular basis of PI activity and relevance to the treatment of breast cancer, incorporating in vitro mechanistic studies with genomic and protein-level data from large human clinical sample cohorts.

# Results and Discussion

### Basal-like breast cancer cell lines are dependent on i-proteasome activity

We initially characterized the relationship between proteasome subunit expression and sensitivity to chemical or genetic inhibition of the proteasome in vitro. Analysis of breast cancer cell line gene expression data (Neve et al, 2006) indicated significantly higher expression of

i-relative to constitutive subunit in basal-A and claudin-low (basal-B) lines than luminal and HER2+ cell lines (Fig 1A and B). We then performed bortezomib and carfilzomib dose–response assays with a panel of lines that model a range of proteasome subunit levels. Expression of inducible relative to constitutive subunit RNA was anti-correlated with response to both inhibitors (Figs 1C and S1A). The relationship to PI sensitivity was most noticeable for *PSMB8* (bortezomib $LD_{50}$ correlation $r$ − 0.76, $P = 0.01$), which was also evident at the protein level (Fig 1D). There was no obvious relationship between bortezomib sensitivity and baseline expression of the proteasome activator cap subunit PA28γ (Fig 1D). This was not a consequence of preferential i-proteasome inhibition, as the total proteasome activity was suppressed in both MDA-MB-468 (i-prot$^{high}$, low bortezomib $LD_{50}$); and MCF-7 (i-prot$^{low}$, low bortezomib $LD_{50}$) (Fig 1E and F).

To rule out off-target effects, we transfected six lines with *PSMB8*-targeted siRNAs and measured cell viability after 48 h using flow cytometry. With the exception of MDA-MB-231, i-prot$^{high}$ basal lines were significantly more sensitive to *PSMB8* depletion than luminal lines (Fig 1Gi). MDA-MB-468 (basal/i-prot$^{high}$) and MCF7 (luminal/i-prot$^{low}$) transfected with *PSMB5*-targeted siRNAs showed no significant differences (Fig 1Gii, with confirmation of RNA and protein knockdown in Fig S1B and C).

### Bortezomib sensitivity correlates with the UPR in vitro

We reasoned that in cell lines addicted to the i-proteasome, pharmacologic suppression of proteasome activity would cause ER stress and trigger the UPR. Three parallel signaling axes mediate the UPR, initiated by ER-membrane sensors: PRKR-like ER kinase (PERK), activating transcription factor 6α (ATF6α) and inositol-requiring protein 1α (IRE1α). Induction of ATF4 marks PERK activation, whereas the IRE1α branch leads to expression of a specific X-box transcription factor splice isoform, XBP1s, which is associated with UPR-related effects in the nucleus. Collectively, these pathways induce genes required for protein folding, secretion, and clearance, or apoptosis if misfolding cannot be resolved (Fig 2A [Wang & Kaufman, 2014]).

In line with our hypothesis, bortezomib treatment induced *ATF4* in the four lines assayed (Fig 2B). Further analysis of i-proteasome–dependent MDA-MB-468 cells revealed induction of XBP1s within 24 h of treatment, concomitant with apoptosis markers and reduction of pro-survival marker NFκB (Fig 2C). Thus, *PSMB8*-high, i-proteasome–addicted BC cell lines exhibit UPR-driven apoptosis in response to proteasome inhibition (Fig 2D). In terms of extrinsic modulation, IFN-γ–induced *PSMB8* in both MDA-MB-468 and MCF7 (Fig 2Ei), but there was no effect on bortezomib sensitivity (Fig 2Eii). Assaying key IFN-γ targets after treatment, including the IFN-stimulated response element transactivator *IRF1*, confirmed the IFN-γ-IRF1-STAT1-PSMB8 axis was intact (Fig 2F). Moreover, siRNA-mediated *IRF1* depletion suppressed target induction (Fig 2F). Thus, although i-proteasome levels are associated with bortezomib sensitivity, this is not an exclusive determinant. Sensitivity is likely to be linked to functional dependence on the i-proteasome rather than the expression of its components.

### i-proteasome induction is associated with the UPR in TNBC

To explore whether PI therapy is an appropriate strategy for treatment of breast cancer, we mined several large genomic

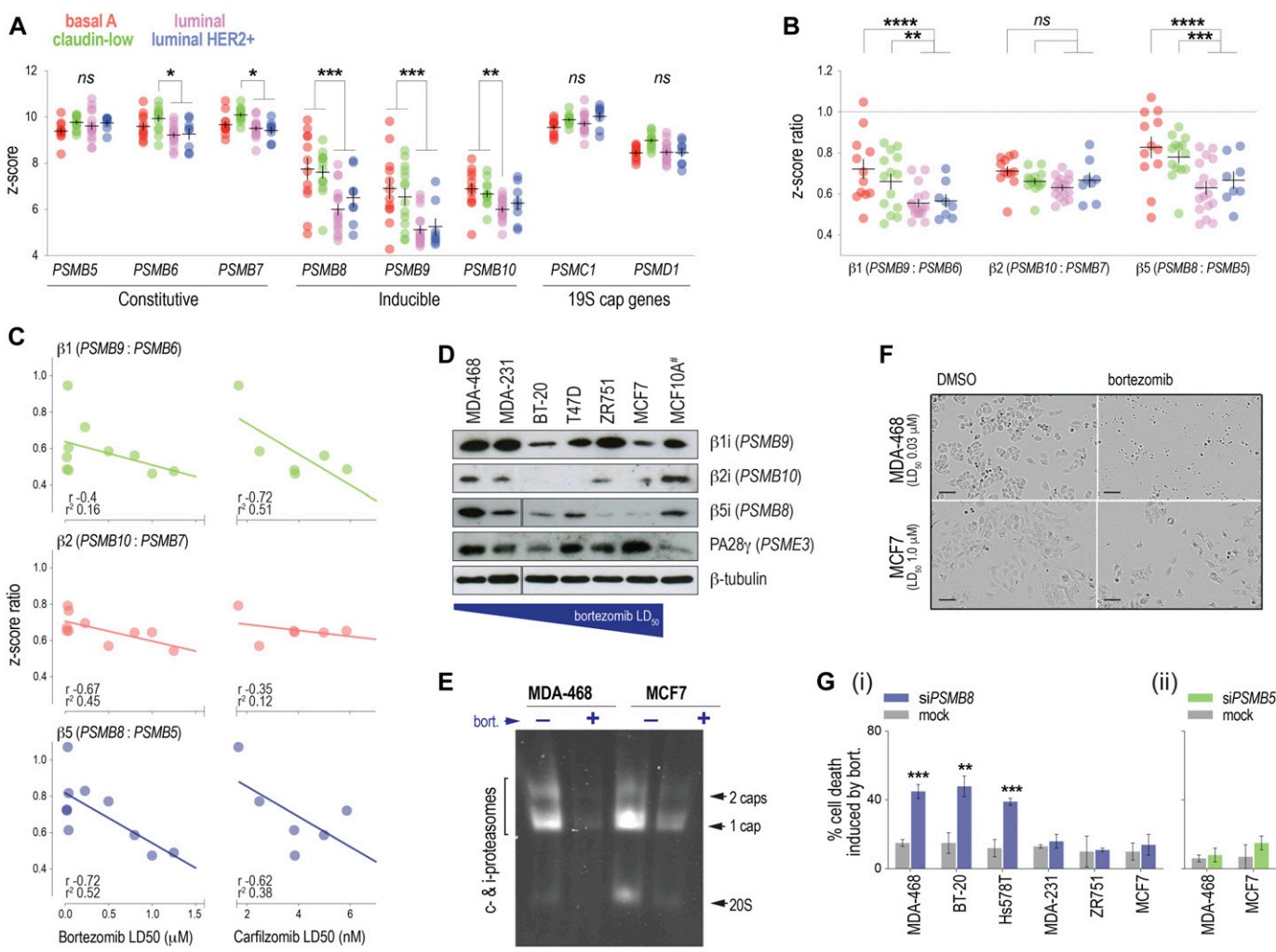

**Figure 1. In vitro analysis of proteasome subunit expression and sensitivity to proteasome inhibitors.**
**(A)** Proteasome subunit expression in breast cancer cell lines (Neve et al, 2006). **(B)** Inducible-to-constitutive subunit expression ratios (i:c) across molecular breast cancer subtypes (stats: two-way ANOVA). **(C)** Linear regression and Pearson correlation analysis of the relationships between i:c and sensitivity to bortezomib or carfilzomib (LD50, lethal dose 50%). Correlation coefficients (r) and regression fit values ($r^2$) indicated. **(D)** Western analysis of inducible subunits and the PA28 cap in lines with a range of bortezomib sensitivities. #MCF10A is a spontaneously immortalized breast-derived line included for comparison. **(E)** Native in-gel proteasome activity assay with lysates from MDA-MB-468 and MCF7 with/without 2-h bortezomib treatment. **(F)** Light microscope images of MDA-MB-468 and MCF7 48 h after bortezomib treatment (captured at 20× magnification, scale bar 50 μm). **(G)** Cell viability after siRNA-mediated depletion of *PSMB8* (i) or *PSMB5* (ii). *P*-values in this figure were from unpaired, two-tailed *t* tests (pair-wise comparisons) or one-way ANOVA tests (comparison across multiple groups): *P < 0.05; **P < 0.01; ***P < 0.001; ****P < 0.0001.
Source data are available for this figure.

datasets: The Cancer Genome Atlas (TCGA; RNA-sequencing data from 1,092 cases including 176 TNBCs) (Cancer Genome Atlas Network, 2012), the Molecular Taxonomy of Breast Cancer International Consortium (METABRIC; expression array data from 1,980 cases incl. 333 TNBCs) (Curtis et al, 2012), and KM-Plotter, a composite of expression array studies with comprehensive clinical annotation (Gyorffy et al, 2010). We interrogated these datasets to identify patient subgroups that may benefit from PIs and to test the hypothesis that i-proteasome activation is associated with the UPR in human tumors. Meta-analysis of expression data indicated that compared with estrogen receptor-positive (ER+) and HER2+ tumors, TNBCs preferentially express i-proteasome subunit genes over constitutive counterparts (β1i:β1, β2i:β2, and β5i:β5;

Fig 3A), and this is most evident in tumors with a claudin-low phenotype (Fig 3B).

We then performed UPR gene set enrichment analysis (The Gene Ontology Consortium, 2019) after ranking the transcriptome according to correlations with *PSMB8*. The distributions of multiple UPR gene sets were skewed toward genes co-expressed with *PSMB8* (Fig 3C). This was evident in both datasets, specifically in TN, basal-like tumors. *PSMB9* and *PSMB10* were among the top 20 *PSMB8*-correlated genes (r 0.89/0.83 and 0.68/0.54 in TCGA/META TNBCs), reflecting coordinated co-expression of i-proteasome subunits. These findings support the idea that i-proteasome addiction could be a therapeutically targetable vulnerability in breast cancer.

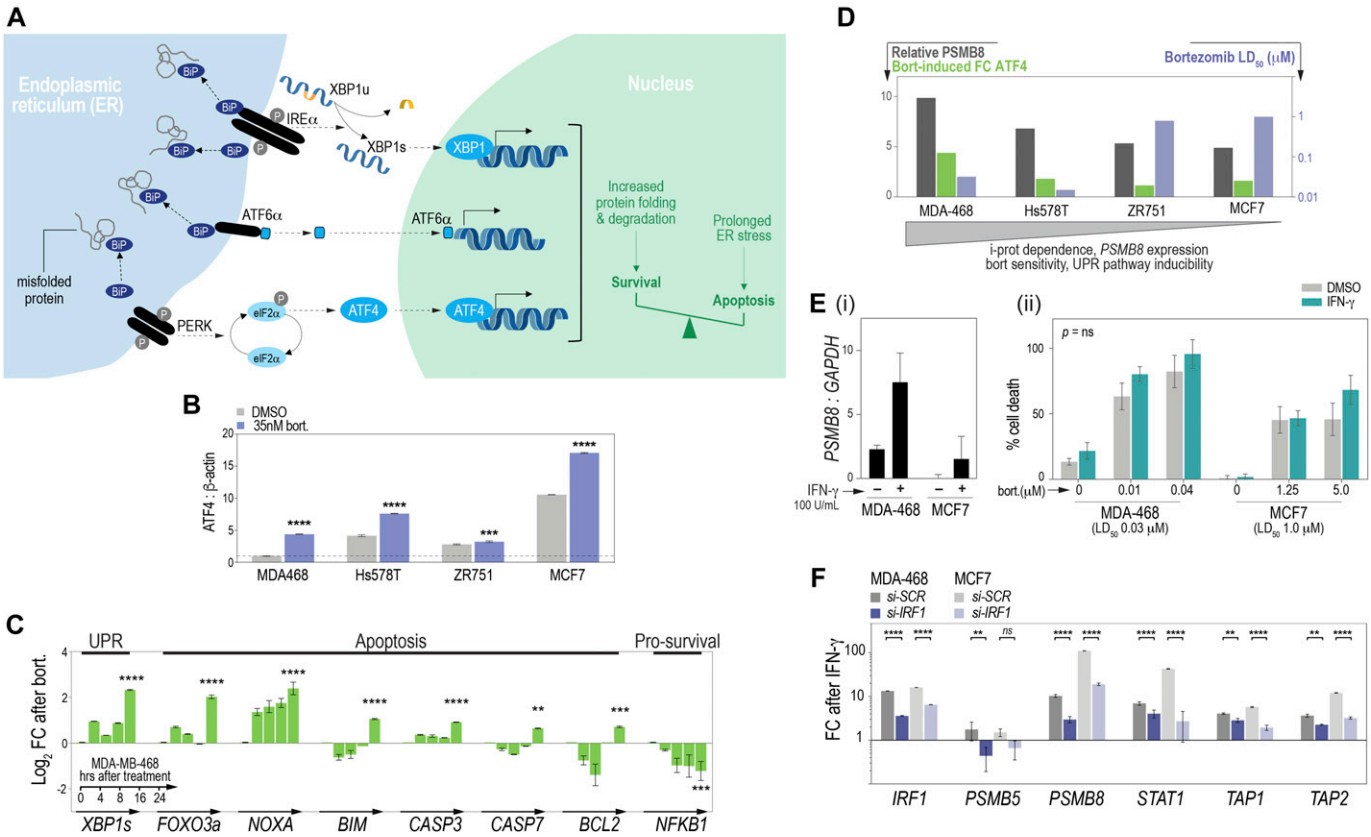

**Figure 2. In vitro activation of the unfolded protein response (UPR) by bortezomib in relation to proteasome subunit expression.**
**(A)** Working model of UPR induction by three major signaling axes. BiP, binding immunoglobulin protein (GRP78). **(B)** qRT-PCR for *ATF4* following bortezomib treatment.
**(B, C)** Log$_2$ fold-change (FC) in *XBP1s*, apoptosis markers, and *NFκB* at multiple time points after bortezomib treatment (B, C: qRT-PCR). **(D)** Inverse association between *PSMB8*/UPR induction and bortezomib sensitivity. **(E)** *PSMB8* qRT-PCR (i) and bortezomib-induced cell death (ii) after pretreatment with IFN-γ. **(F)** IFN-γ–mediated induction of *IRF1* and antigen processing genes in MDA-MB-468/MCF7 cells transfected with scrambled (SCR) or *IRF1*-specific siRNAs. *P*-values in this figure were from unpaired, two-tailed *t* tests: **$P < 0.01$; ***$P < 0.001$; ****$P < 0.0001$.

## i-proteasome switching is driven by gene copy number alterations (CNAs) and stratifies survival in TNBC

*PSMB8* and *PSMB9* genes are located within a 25-Mbp region on chromosome 6p that is frequently affected by CNAs in cancer, specifically, in the class-II locus of the HLA complex, which spans 4 Mbp on chromosome 6p.21 (Fig 4A). 6p gains are common in high-grade malignancies (Santos et al, 2007), suggesting proteotoxic stress resistance may be a fundamental requirement for cancers with high metabolic activity. TCGA CNA data indicated that 6p is affected by complex genomic instability in TNBC, marked by focal gains and losses (Fig 4Ai). *PSMB8/9* are gained in 46% and amplified in 5.2% of TNBCs. The locus is also gained in ER+ and HER2+ cases, but at lower frequency (17% and 27%; Fig 4Aii). Flanking *PSMB8/9* are co-altered genes encoding *TAP1* and *TAP2*–ER membrane channels that internalize proteasomal peptides and facilitate MHC class-I antigen presentation (Fig 4Aii).

Quantifying proteasome subunit gene CNAs across BC sub-types revealed that in addition to gains at 6p21.32 (*PSMB8/9*) in TNBC, other events frequently affecting proteasome subunit CN are losses at 17p13.2 and 16q22.1 (*PSMB6*, *PSMB10*; Fig 4B). However, the genomic landscape of TNBC is unique in that

i-subunit gains occur concomitantly with loss of constitutive subunit genes in more than half of cases (Fig 4B). This was also evident within individual cases, with 65.3%, 31.2%, and 58.4% of TNBCs exhibiting a predominance of inducible *β1*, *β2*, and *β5* subunits, respectively (Fig 4C). Considering that subunit gene copy number is a determinant of overall expression (Fig S3A), these findings suggest that high i-proteasome expression in TNBC is at least partly a consequence of selection pressure during tumorigenesis.

Next, we used Kaplan–Meier analysis to investigate associations between proteasome subunit expression and patient outcomes. In independent TNBC datasets, i:c subunit expression ratios stratified 10-yr breast cancer–specific survival, but cases with the *highest* i:c ratios had longer survival (Figs 4D and S2). Stratification was more prominent among patients treated with chemotherapy and/or radiotherapy (Figs 4D and S2). Therefore, althoguh i-proteasome switching subverts metabolic stress in vitro and may confer a selective advantage during TNBC development, in the clinical setting, this seemed to be associated with favorable responses to treatment.

Because (1) the i-proteasome is associated with both metabolic homeostasis and antigen processing (2), multiple antigen-processing

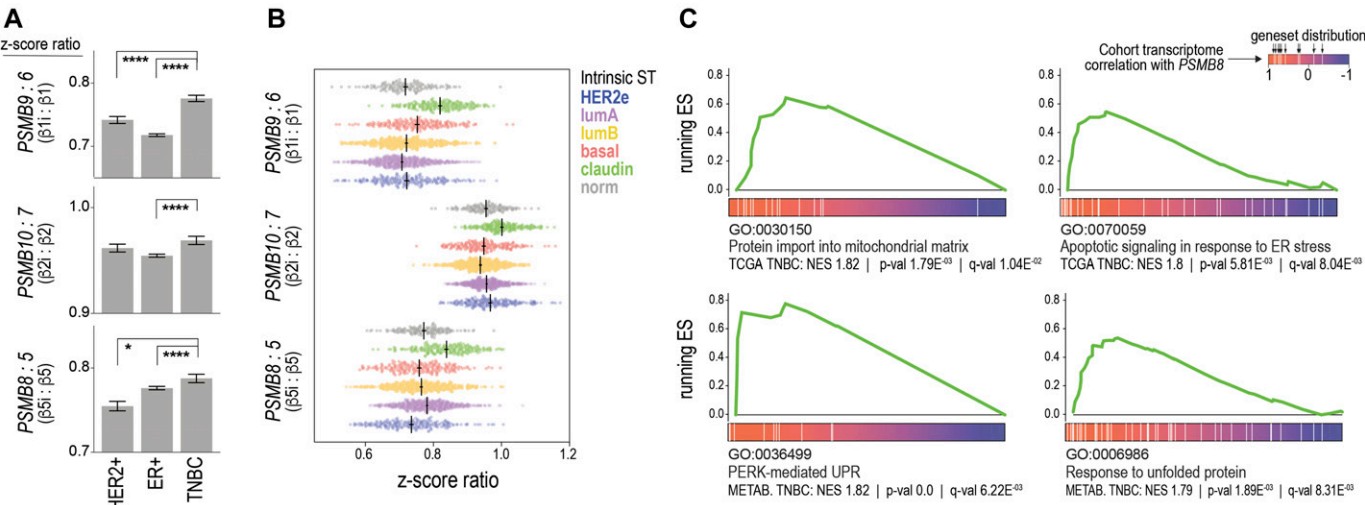

**Figure 3.  i-proteasome genes are induced in claudin-low TNBCs and coordinately expressed with unfolded protein response genes.**
**(A, B)** i:c Subunit expression ratios in breast cancer subtypes (METABRIC data, ANOVA test. *$P$ < 0.05; ****$P$ < 0.0001). **(C)** Gene set enrichment analysis plots showing running enrichment scores (ESs) skewed toward the *PSMB8*-correlated transcriptome. NES, normalized ES.

machinery components are co-altered by genomic alterations in TNBC, and (3) efficient antigen processing is conducive to anti-tumor immunity in cancer generally; we reasoned that i-proteasome overexpression could be both a vestige of metabolic addiction *and* indicator of effective antigen presentation. The fact that the link between i-proteasome and clinical outcome is specific to patients treated with chemotherapy, and radiotherapy is consistent with the evidence that these treatments promote immunogenic tumor cell death by reactivating immune surveillance (Dushyanthen et al, 2015). Indeed, i-subunit but not c-subunit expression is inversely proportional to tumor purity (Fig S3B), and the i:c expression ratio is highest in "basal-like, immune-activated" TNBCs (Fig 4E) characterized by dense lymphocytic infiltrates (Burstein et al, 2015).

This goes some way to reconciling the i-proteasome's involvement with both tumor development and responses to treatment, but also raises the question of how much of the i-proteasome RNA measured in tumor homogenate is attributable to tumor versus stromal components. Unlike RNA expression, CNAs are intrinsic to the tumor cell compartment, so we performed Kaplan–Meier analysis of TNBCs after classifying them according to shifts in inducible and constitutive proteasome subunit gene copy number. In both METABRIC and TCGA datasets, TNBCs with higher overall i-subunit copy number survived longer after the treatment (Fig 4F), supporting the idea that higher levels of inducible proteasome subunits are protective.

### β5i is a favorable prognostic indicator associated with tumor-specific immunity in TNBC

To validate these findings at the protein level, we performed immunohistochemistry (IHC) analysis of β5i (*PSMB8*) in relation to clinicopathologic variables and disease-specific survival using a third cohort of 424 invasive breast tumors. We also analyzed the

MHC–I complex and proteasome activator subunit PA28β (*PSME2*), which is similarly induced by IFN-γ but located outside the HLA complex on chromosome 14q12 and less frequently affected by CNAs (14.5% and 1.7% of TNBCs with *PSME2* gain or amplification; 46% and 5.2% for *PSMB8*). β5i and PA28β were detected in the nuclei and cytoplasm of normal mammary epithelial structures and breast tumor cells, and MHC-I in cytoplasm and cell membranes (Fig 5A and Tables S1–S3). On average, 83% of cases were strongly positive for β5i, and 17% showed selective loss in the tumor compartment (Figs 5A and S4A). Luminal-A–like tumors had the most frequent expression overall, followed by basal-like TNBCs (Fig S4B). Around one-third of all tumors exhibited loss of MHC-I expression, but again expression was frequently maintained in basal-like TNBCs (Fig S4C and D and Table S2). For PA28β, 92% of cases were positive, with minimal differences between breast cancer subtypes (Fig S4E and F and Table S3). Among TNBCs, there was a direct relationship between levels of β5i and PA28β, and they were both strongly associated with expression of MHC-I (Fig 5B), indicating that overall, there is coordinated expression of MHC-I antigen processing and presentation pathway components in TNBC.

We then reviewed matching hematoxylin and eosin–stained whole sections to quantify immunologic correlates of β5i, PA28β, and MHC-I expression: (1) the density of stromal TILs (Salgado et al, 2015), an established marker of treatment response in TNBC (Fig 5Ci), (2) the frequency of PD-L1 expression by TILs (Fig 5Cii), and (3) tumor cells (Fig 5Ciii), a marker of effector T-cell responses in breast cancer (Dushyanthen et al, 2015), and (4) the density of stromal cells expressing high levels of MHC-II, a marker of professional APCs (macrophages, dendritic cells, and B-cells; Fig 5Civ). Contingency analysis showed that TNBCs with sustained expression of β5i and MHC-I were more likely to be infiltrated by TILs and APCs and to exhibit signs of antigen-specific T-cell engagement. These findings are consistent with another report showing a direct relationship

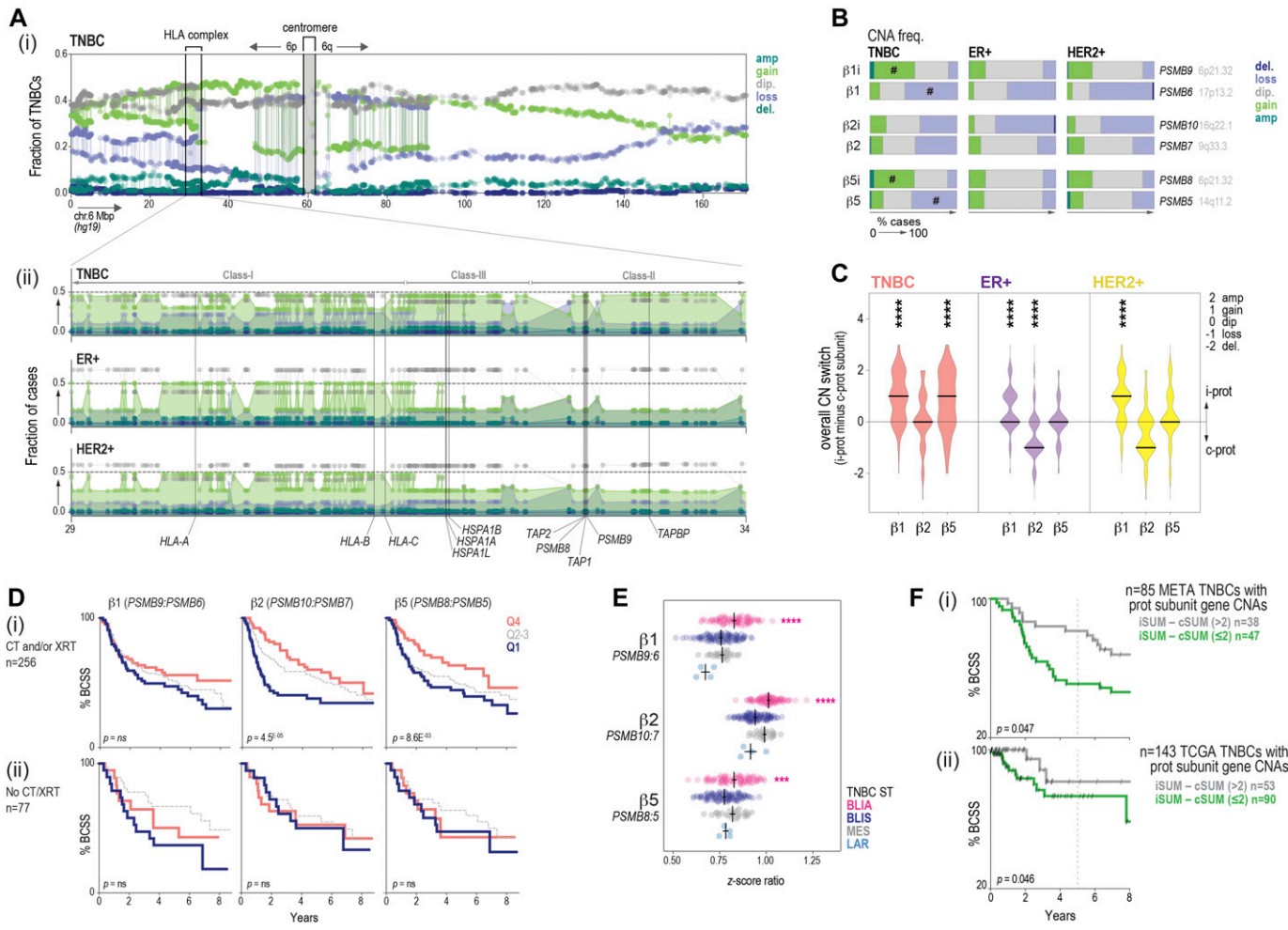

**Figure 4. Proteasome gene copy number aberrations related to clinical outcomes in breast cancer.**
**(A)** TCGA 450k methylation array data for chromosome 6. (i) GISTIC calls for each probe shown as fractions of all triple-negative breast cancers (TNBCs). (ii) Zoomed region encoding antigen-processing genes. Fractions of HER2+ and ER+ cases shown for comparison. **(B)** CN status of subunit genes in major disease subtypes. #, instances of inducible subunit gain and constitutive subunit loss in a large percentage of TNBCs. **(C)** Violin plots showing i:c subunit CN switching (stats: pairwise Kruskal–Wallis tests with Dunn's correction for multiple comparisons; ****$P$ < 0.0001). **(D)** Kaplan–Meier analysis of i:c subunit expression in TNBC patients treated with (i) or without (ii) chemotherapy (CT) and/or radiotherapy (XRT). Q4/2-3/1, upper/mid/lower quartiles. Log-rank $P$-values shown. **(E)** i:c subunit expression ratios in TNBC subtypes (Burstein et al, 2015): BLIA, basal-like immune-activated; BLIS, basal-like immune-suppressed; LAR, luminal androgen receptor-like; MES, mesenchymal. Kruskal–Wallis test: ***$P$ < 0.001, ****$P$ < 0.0001. **(F)** Kaplan–Meier analysis of (i) METABRIC and (ii) TCGA TNBCs classified by whether i-subunit gene copy number outnumbers that of constitutive subunit counterparts. Log-rank $P$-values shown.

between tumor β5i, ER stress markers, MHC-I, and TILs (Lee et al, 2019).

To investigate the potential consequences of i-proteasome components and/or MHC-I being dysregulated or lost in TNBC, we performed the Kaplan–Meier analysis. Expression of β5i or MHC-I modestly stratified survival (Fig 5Di, Ei, and Fi), but more striking differences began to emerge when we considered the prognostic significance of TILs in the tumor microenvironment of TNBCs that maintained or lost antigen presentation components. TIL density remains the most reliable prognostic indicator in TNBC (Loi et al, 2019; Luen et al, 2019), yet we found that this was specific to tumors that retain expression of β5i, with little difference in outcome over 20 yr among TNBCs that had lost β5i expression (Fig 5Dii). MHC-I and PA28β showed similar trends, although their interactions with TILs density were not as significant (Fig 5Eii and

Fii). We reasoned that if β5i loss enabled immune escape, expression would be suppressed during metastatic progression. Indeed, IHC analysis of β5i in brain metastases compared with patient-matched primary breast tumor samples revealed lower β5i expression in brain metastases in 16/34 of cases (Fig 5G; paired $t$ test $P$ = 0.007).

Taken together, these findings suggest that antigen processing via i-proteasome subunit β5i is associated with antitumor immune responses. Given that we saw such a striking difference in TIL-based prognostication depending on the β5i status, further cohort studies are warranted to investigate predictive testing of i-proteasome expression in conjunction with TILs assessment. We explored the possibility that β5i-high TNBCs might be intrinsically more immunogenic by comparing the mutation burden of *PSMB8* gained versus copy number–neutral TNBCs in the TCGA cohort but found no evidence

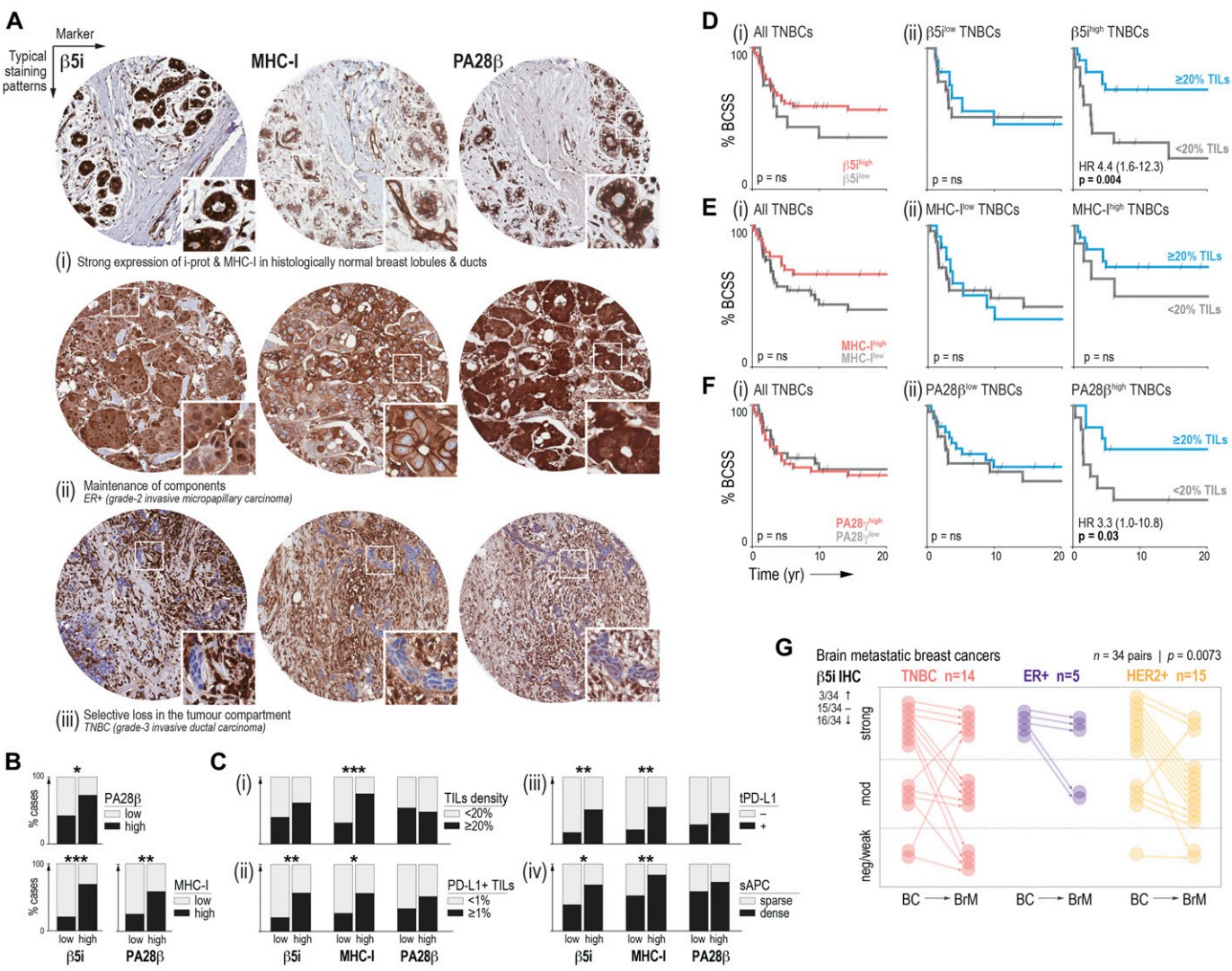

**Figure 5. Tumor compartment–specific expression of PSMB8 relates to clinicopathologic variables.**
**(A)** Representative β5i, MHC-I, and PA28β IHC analysis of normal breast ducts and lobules (i) and invasive breast tumors (ii, iii), illustrating cases that exemplify maintenance (ii) or selective loss (iii) of these class-I antigen presentation components. Cores are 1.0 mm i.d. and insets 140 $\mu m^2$. **(B)** Contingency analysis of the relationships between β5i, MHC-I, and PA28β in triple-negative breast cancer (TNBC). Fisher's exact test *P*-values indicated *$P < 0.05$; **$P < 0.01$; ***$P < 0.001$. **(C)** Contingency analysis of relationships between β5i, MHC-I, or PA28β, with TILs density (i), TILs PD-L1 positivity (ii), tumor cell PD-L1 positivity (iii), and the density of stromal APCs (sAPCs; iv). Fisher's exact test *P*-values indicated *$P < 0.05$; **$P < 0.01$; ***$P < 0.001$. **(D, E, F)** Kaplan–Meier analysis of TNBCs stratified by β5i, MHC-I, or PA28β (i) or by TIL density after classifying TNBCs by their maintenance or loss of β5i, MHC-I, or PA28β (ii). HR, hazard ratio (95% confidence interval); log-rank *P*-values shown. **(G)** Change in β5i IHC scores in brain metastases (BrM) compared with matching primary breast cancers (BC). Overall numbers of cases exhibiting increases (↑), decreases (↓), or no change (−) are indicated. Paired, two-tailed *t* test *P*-value shown.

suggesting that their neoantigen load is higher than other TNBCs (Fig S4G and H). Hence, β5i expression is unlikely to be causally associated with a dense TIL phenotype, but probably potentiates tumor-specific immune responses providing there is sufficient capacity.

We propose that the i-proteasome is exploited during TNBC development to cope with proteotoxicity, but in immunologically "hot" tumors (defined here as TILs occupying ≥20% of tumor-associated stroma), this ultimately becomes a liability because i-proteasome activity is linked to the efficacy of radiotherapy and chemotherapy (Fig 6). Hence, adjuvant PI therapy could be counterproductive in TNBC. On the other hand, *increasing* i-proteasome activity could potentiate other first-line therapies.

Supporting this idea in principle, Tripathi and colleagues showed that *PSMB8/9* are suppressed in mesenchymal-like non–small cell lung cancers, and that reactivating these genes with exogenous IFN-γ restored a repertoire of MHC-I–bound tumor antigens that could efficiently prime cytotoxic responses from patient-derived peripheral blood mononuclear cells (Tripathi et al, 2016). 5-Aza-2′-deoxycytidine (decitabine), a cytosine analog and DNA methylation inhibitor, had the same effect by suppressing methylation at *PSMB8/9* enhancers (Tripathi et al, 2016).

Considering these findings, our study supports the rationale for trialing immunotherapy and DNA methyltransferase inhibitor combinations in TNBC and other solid cancers (Jones et al, 2019).

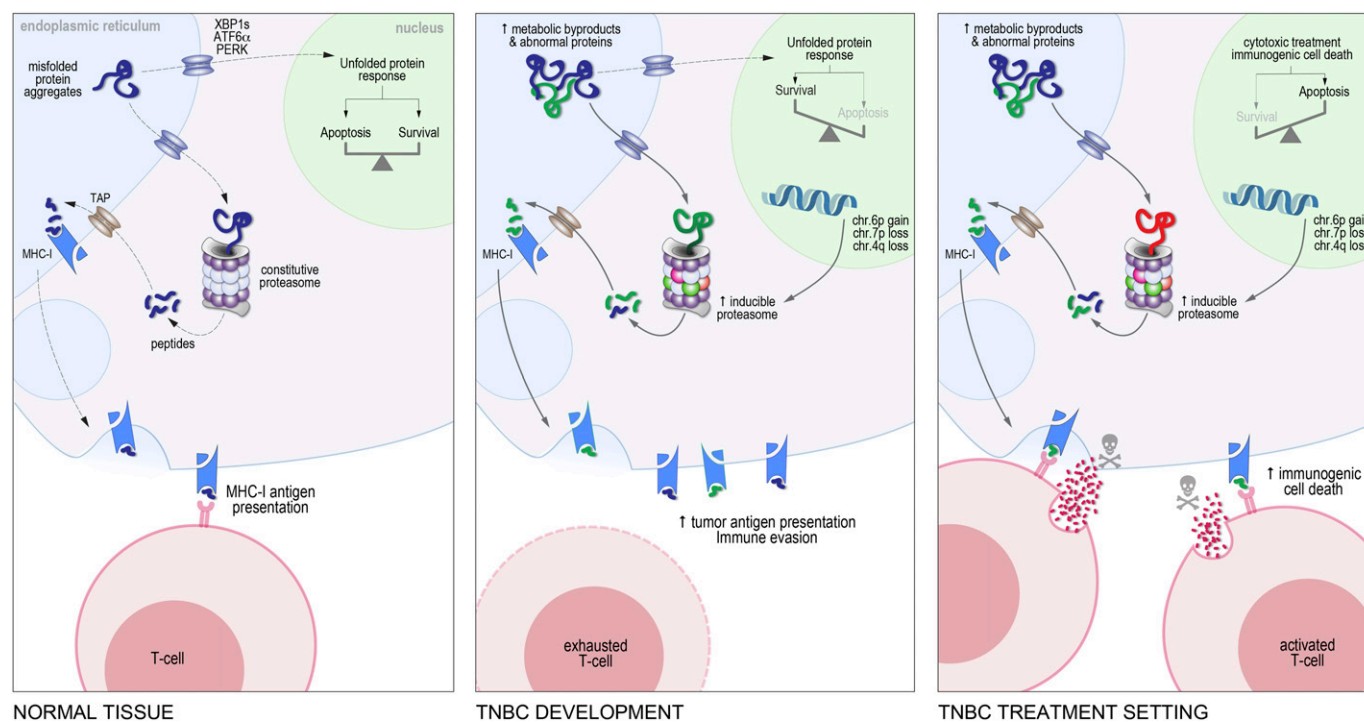

**Figure 6.  Model contrasting the potential consequences of i-proteasome expression before and after treatment.**

Moreover, we recommend that additional cohort studies are warranted to quantify the prognostic and predictive value of adding β5i IHC to diagnostic assessment of TILs density in TNBC.

# Materials and Methods

### Cell lines and qRT-PCR

Cell lines were purchased from the American Type Culture Collection, maintained in recommended culture conditions and regularly screened for Mycoplasma (MycoAlert; Lonza). Cell line working stocks used in this study were authenticated by STR profiling (Promega GenePrint 10 System). For RNA inhibition studies, the cells were transfected in six-well plates ($3 × 10^5$ cells/well) with 50 nM siRNAs (GenePharma) using Lipofectamine RNAiMAX (Life Technologies). An IncuCyte live-cell imaging system (Essen BioScience) was routinely used to monitor cell attachment and growth in real time. Expression analyses were performed 72 h post-transfection. For qRT-PCR, total RNA was extracted from cultured cells using the RNeasy Mini Kit (QIAGEN), cDNA was random-primed from 1 μg total RNA (MMLV RT; Life Technologies), and qPCR was performed using SYBR Green (Life Technologies; Table 1) and a CFX96 instrument (Bio-Rad). Experiments were repeated three times.

### Western analysis

Cells were trypsinized, centrifuged, washed, and then lysed in Hepes (100 mM; pH 7.5), EDTA (2 mM), NaF (100 mM), sodium chloride (500 mM), and trypsin inhibitor (50 μg/ml; Promega) with fresh

cOmplete Mini Protease Inhibitor (Roche). Protein lysates (5–20 μg) were resolved by SDS–PAGE and probed overnight at 4°C with antibodies against PSMB5 (#12919), PSMB6 (#13267), PSMB7 (#13207), PSMB8 (#13726), and PA28g (#2142S) from Cell Signaling, used at 1:1,000, or β-tubulin (SAB4700544, 1:200; Merck). Chemiluminescent detection was carried out using appropriate secondary antibodies conjugated to horseradish peroxidase and the enhanced chemiluminescence kit (Amersham).

### Native in-gel proteasome activity assay

Chymotrypsin-like activity was measured in freshly prepared cell lysates as described previously (Elsasser et al, 2005) with minor modifications. In brief, $5 × 10^6$ cells were washed three times with ice-cold PBS, pelleted by centrifugation (5 min, 250$g$ at 4°C), and then resuspended in lysis buffer (10 mM Tris–HCl [pH 7.8], 5 mM ATP, and 5 mM MgCl$_2$) and kept on ice for 10 min. The cells were sonicated (MSE ultrasonic disintegrator, amplitude 15, 10 s at 4°C) followed by centrifugation to remove cell debris (5 min, 16,000$g$, 4°C). Whole cell lysates (40 μl) were separated on 3.5% non-denaturing polyacrylamide gels (Bio-Rad) in 10 mM Tris–HCl buffer supplemented with ATP (0.5 mM; Sigma-Aldrich), MgCl$_2$ (5 mM), glycerol (10% vol/vol), and DTT (0.5 mM). Electrophoresis was performed at 35 V for 30 min at 4°C, then the voltage was increased to 75 V for 4 h. Peptidolytic activity was detected by incubating gels in Suc-LLVY-MCAc substrate (dissolved in 50 mM Tris [pH 8.0], 5 mM MgCl$_2$, 1 mM DTT, 2 mM ATP, and 0.02% SDS for 10 min at 37°C). Proteasome bands were identified by the release of highly fluorescent, free 7-amino-3-methylcoumarin (AMC) under UV light (ChemiDoc; Bio-Rad).

**Table 1.  siRNA and primer sequences.**

| siRNAs | | |
|---|---|---|
| PSMB8 | CCACUCACAGAGACAGCUAUU | |
| IRF1 | GAAAGUUGGCCUUCCACGUCU | |
| PSMB5 | AAGCUCAUAGAUUCGACAUUG | |
| Non-targeted negative control | UUCUCCGAACGUGUCACGUTT | |
| Primers | | |
| PSMB6 | CAAGCTGACACCTATTCACGAC | CGGTATCGGTAACACATCTCCT |
| PSMB7 | ATCGCTGGGGTGGTCTATAAG | AAGAAATGAGCTGGTTGTCAT |
| FOXO3 | TCTTCAGGTCCTCCTGTTCCTG | GGAAGCACCAAAGAAGAGAGAAG |
| NOXA | AGAGCTGGAAGTCGAGTGT | GCACCTTCACATTCCTCTC |
| BIM | GTATTCGGTTCGCTGCGTTC | GCGTTTCTCAGTCCGAGAGT |
| CASP3 | TGCTATTGTGAGGCGGTTGT | TCACGGCCTGGGATTTCAAG |
| CASP 7 | GTGGGAACGATGGCAGATGA | GAGGGACGGTACAAACGAGG |
| BCL2 | GTGAAGTCAACATGCCTGCC | ACAGCCTGCAGCTTTGTTTC |
| NFKB | CGCGCCGCTTAGGAGGGAGA | GGGCCATCTGCTGTTGGCAGT |
| PSMB5 | CCGCGCTCTACCTTACCTACCT | GCATGGCTTAATCTTTGAGACAAG |
| PSMB8 | CGTCACCAACTGGGACGACA | CTTCTCGCGGTTGGCCTTGG |
| IRF1 | AGCTCAGCTGTGCGAGTGTA | TAGCTGCTGTGGTCATCAGG |
| STAT1 | CGGGCTCCTTCTTCGGATTC | CAGAGGTAGACAGCACCACC |
| XBP1 | TCCTGTTGGGCATTCTGGAC | GGCTGGTAAGGAACTGGGTC |
| TAP1 | TAGTCTGGGCAGGCCACTTT | CTCGGAAAGTCCCAGGAACA |
| TAP2 | AGTGCTGGTGATTGCTCACA | GAACCAGGCGGGAATAGAGG |
| ATF4 | CTTGATGTCCCCCTTCGACC | GAAGGCATCCTCCTTGCTGT |

### In vitro bortezomib/carfilzomib sensitivity assays

A panel of basal-A (BT20 and MDA-MB-468), basal-B (Hs578T and MDA-MB-231) luminal-like (BT-483, MCF7, and T47D), and luminal/HER2+ (SKBr3 and ZR751) cell lines were used for cytotoxicity experiments. The cells were routinely seeded in 96-well plates ($2 \times 10^4$ cells/well) with varying concentrations of PIs, harvested after 48 h, centrifuged at $1,300g$, washed in PBS, and stained with 7-amino-actinomycin-D solution (7AAD, 2 µg/ml; Invitrogen) for 10 min at room temperature. Cell viability was determined with a FACSCalibur flow cytometer (Becton Dickinson) and analyzed with FLOWJO software (Tree Star Inc.). Cell survival data were normalized (0–100% defined as minimum and maximum values), and regression analysis was performed using GraphPad Prism software (v8.4) to define LD50 values.

### Datasets and statistics

The following datasets were used in this study: (1) Affymetrix HG-U133A gene expression array data: n = 51 breast cancer cell lines (Neve et al, 2006), (2) RNASeq (V2 RSEM) mRNA expression z-scores: n = 1,108, provisional TCGA breast tumor dataset (Cancer Genome Atlas Network, 2012), (3) GISTIC 2.0 putative copy number calls: n = 1,080, provisional TCGA breast tumor dataset (Cancer Genome Atlas Network, 2012), (4) Illumina HT-12 gene expression array data: n = 1,980, METABRIC invasive breast cancer dataset (Curtis et al, 2012), (5)

mRNA expression array data: n = 256 TNBCs (IHC classification) and n = 400 basal-like tumors (PAM50 classifier) from KM plotter for breast cancer (Gyorffy et al, 2010), and (6) computationally derived pan-cancer tumor purity assessment of TCGA samples (Aran et al, 2015).

Gene set enrichment analysis was performed using the GSEAPreranked module of GenePattern (weighted scoring scheme, 1,000 permutations) (Reich et al, 2006). Gene sets were extracted from the Gene Ontology Consortium database (The Gene Ontology Consortium, 2019) using the search term "UPR." Preparation of graphs and all other statistical tests were performed using GraphPad Prism software (v8.4). Statistical tests are described in figure legends.

### Clinical sample cohorts and IHC analyses

IHC analysis of archival specimens and clinicopathologic data were approved by human research ethics committees at The Royal Brisbane and Women's Hospital (2005-022) and The University of Queensland (2005000785). IHC analyses were performed on two separate cohorts:

1.  The Queensland follow-up cohort. This resource was built from archival tissue specimens of breast cancer patients treated in Queensland between 1987 and 1992, sourced from the statewide Pathology Service, Pathology Queensland, and sampled in duplicate on tissue microarrays (TMAs). Median follow-up is 13.9 yr (range

**Table 2. Immunohistochemistry details.**

| Target | Antigen retrieval[a] | Primary Ab block | Ab manufacturer and clone | Primary Ab[b] | Scoring[c] |
|---|---|---|---|---|---|
| $\beta$5i (PSMB8) | Citrate buffer 100°C 20 min | Background sniper BSA 1% | Cell Signaling #13726 Mouse IgG1, IA5 | 1:400 1.5 h, RT | Tumor |
| PD-L1 | EDTA buffer 95°C 60 min | Background sniper | Cell Signaling #13684 Rabbit IgG, E1L3N | 1:200 2 h, RT | sTILs, tumor |
| MHC-II (HLA-DP/Q/R) | Citrate buffer 121°C 5 min | Goat serum 10% BSA 1% | Abcam #ab86261 Mouse IgG1, KUL/05 | 1:100 4°C RT | Stroma |
| MHC-I (HLA-A/B/C) | Dako retrieval buffer pH 6.0 100°C 10 min | Background sniper BSA 1% | Abcam #70328 Mouse IgG1, EMR8-5 | 1:800 1 h, RT | Tumor |
| PA28 (PSME2) | None | Background sniper BSA 1% | Abcam #ab183727 Rabbit IgG, EPR14931 | 1:1,000 1 h, RT | Tumor |

[a]Citrate buffer: 0.01M citrate buffer, pH 6.0, EDTA buffer: 0.001M Tris–EDTA, pH 8.8.
[d]Primary antibodies diluted in Da Vinci Green Diluent.
[e]TILs were scored on whole breast tumor sections according to the International Working Group criteria (Salgado et al, 2015). Intensity and percentage of TILs stained were recorded as a Histo-score. A cut-off of ≥1% was considered positive. BSA, bovine serum albumin; ON, overnight; RT, room temperature.

0.3–41 yr), median age at diagnosis 59.3 yr, and there were 199 breast cancer–specific deaths at the 25-yr censor point (~37.7%).

2. A second cohort of brain metastases and patient-matched primary breast tumor specimens collected between 2001 and 2013, also sampled in TMAs.

Clinical and pathology data were extracted from diagnostic reports, our internal diagnostic histopathology review (SRL) and the Queensland Cancer Registry. Analysis included cross-referencing to clinicopathologic parameters that were assessed and published previously (e.g., expression of ER, PR, and HER2; histological grade and subtype [Junankar et al, 2015; Zhang et al, 2015; Burgess et al, 2016; Hernandez-Perez et al, 2017; McCart Reed et al, 2018; Raghavendra et al, 2018; Wiegmans et al, 2019]). For IHC, 4-$\mu$m TMA sections were heat-retrieved in using a Decloaking Chamber and blocked for 15 min at room temperature before staining (Table 2). The MACH 1 Universal HRP-Polymer Detection Kit was used for detection. Reagents and equipment were from Biocare Medical unless otherwise specified. For image analysis, hematoxylin-counterstained sections were scanned at 40× magnification on an Aperio AT Turbo slide scanner (Leica Biosystems). De-identified digital TMA core images were scored by one assessor and reviewed by a second. The maximum scores of duplicate TMA cores were calculated for each case, except for $\beta$5i, MHC-I, and PA28$\beta$, where minimum scores were used because the disease-associated phenotypes were a loss of expression. Associations between biomarkers and clinicopathologic variables were investigated using chi-squared, Fisher's exact, and log-rank tests (GraphPad Prism v8.4).

# Supplementary Information

# Acknowledgements

We are grateful to Clay Winterford (QIMR Berghofer Medical Research Institute), the Brisbane Breast Bank, and patients who donated tissue and clinical information. This study was supported by the Australian National Health and Medical Research Council (APP1113867) and used data from METABRIC, funded by Cancer Research UK and the British Columbia Cancer Agency Branch.

## Author Contributions

A Adwal: conceptualization, data curation, formal analysis, investigation, methodology, and writing—original draft, review, and editing.
P Kalita de-Croft: investigation and writing—review and editing.
R Shakya: investigation and writing—review and editing.
M Lim: investigation and writing—review and editing.
E Kalaw: investigation and writing—review and editing.
LD Taege: investigation and writing—review and editing.
AE McCart Reed: data curation, supervision, investigation, and writing—review and editing.
SR Lakhani: resources, supervision, funding acquisition, investigation, and writing—review and editing.
DF Callen: conceptualization, resources, data curation, supervision, funding acquisition, investigation, project administration, and writing—original draft, review, and editing.
JM Saunus: conceptualization, resources, data curation, formal analysis, supervision, funding acquisition, investigation, visualization, project administration, and writing—original draft, review, and editing.

## Conflict of Interest Statement

The authors declare that they have no conflict of interest.

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
