## [Reviewer comments · Life Science Alliance]

Life Science Alliance

Tradeoff between metabolic i-proteasome addiction & immune evasion in triple-negative breast cancer

Alaknanda Adwal, Priyakshi Kalita de-Croft, Reshma Shakya, Malcolm Lim, Emarene Kalaw, Lucinda Taege, Amy McCart Reed, Sunil Lakhani, David Callen, and Jodi Saunus

DOI: <https://doi.org/10.26508/lsa.201900562>

Corresponding author(s): Jodi Saunus, The University of Queensland and Alaknanda Adwal, The University of Adelaide

Review Timeline:

Submission Date:	2019-09-23
Editorial Decision:	2019-10-30
Revision Received:	2020-04-03
Editorial Decision:	2020-04-22
Revision Received:	2020-05-08
Accepted:	2020-05-12

Scientific Editor: Andrea Leibfried

Transaction Report:

October 30, 2019

Re: Life Science Alliance manuscript #LSA-2019-00562-T

Dr. Jodi M. Saunus
The University of Queensland
Herston

Dear Dr. Saunus,

Thank you for submitting your manuscript entitled "Trade-off between metabolic immunoproteasome addiction and immune evasion in triple-negative breast cancer" to Life Science Alliance. The manuscript was assessed by expert reviewers, whose comments are appended to this letter.

As you will see, the reviewers think that your findings are potentially relevant to the clinics. However, they also think that the current dataset remains somewhat limited and an extension and further cohort analyses are requested (reviewer #1, reviewer #3) as well as testing more specific immunoproteasome inhibitors (rev#2). Reviewer #3 thinks that the conclusion that high expression of the immunoproteasome subunit PSMB8 correlates with survival due to better T cell infiltration is potentially wrong as chemotherapy usually compromises T cell mediated anti-tumor responses. The reviewer suggests an alternative explanation that should get considered in revision as well. Please note that reviewer #2 assessed the work from the immunoproteasome but not clinical angle.

We would thus like to invite you to submit a revised version of your manuscript to us, addressing the reviewer concerns. The proposed additional analyses seem straightforward to add in our view, but please do get in touch in case you would like to discuss individual revision points further.

Thank you for this interesting contribution to Life Science Alliance. We are looking forward to

receiving your revised manuscript.

Sincerely,

B. MANUSCRIPT ORGANIZATION AND FORMATTING:

Reviewer #1 (Comments to the Authors (Required)):

The present study has analyzed the role of the proteasome immunosubunits PSMD8 and IP (immunoproteasome) in the context of TNBC and bortezomib induced proteasome inhibition in vitro, in a clinical setting and by bioinformatic analysis of clinical data sets. They conclude that high IP levels in TNBC correlates with a high density of TILs and thus may be protective. On the other hand high IP level protect cells against toxic effects of proteasome inhibition indicating that proteasome inhibition in a clinical setting may be counterproductive in cases where immune treatment may be of choice.

Although the conclusions of the study are quite interesting from a clinical point of view the results are in general terms neither surprising nor entirely new. It is known that IP are protective against proteotoxic stress and it is also known that high IP levels enhance surface MHC class I expression which is a prerequisite for improved T cell stimulation. Interestingly a very recent study showed that inhibitor resistance (a.o. bortezomib can in part be overcome by application of PSMD8 specific inhibitors).

However, one major criticism I have is that the authors follow a widespread narrow view of IP function by focusing their analysis predominantly on PSMD8 expression thereby neglecting that IP function in the context of proteotoxic stress (inhibitor treatment) is tightly connected with the concomitant expression of PA28 alpha/beta (PSME1/2) and the formation of hybrid IP (19S-20S-PA28) and 20S-PA28 alpha/beta complexes. In addition to these complexes whose formation is induced by IFN γ , also the expression of ERAP1/2 has to be considered in the context to antigen processing. Both PA28ab and ERAP1/2 have been previously reported to be important in anti-tumor responses.

In my view and to add sufficient novelty to the study to justify publication the expression of PA28 ab in the context of inhibitors resistance and the analysis of both PA28ab and ERAP expression in TNBC ought to be included in the study as well to complete the picture.

To do so, a reanalysis of the clinical data sets I would regard as sufficient.

Fig1D:

The figure lacks the expression analysis of standard subunits and upon extension of the study the analysis of PA28ab expression.

Fig1 E:

The quality of the native PAGE analysis is not great. I wonder by which criteria the authors assigned the identity of the individual proteasome complexes. What is needed is a lane with purified 30S/26S complexes and 20S complexes and an immunoblot with antibodies directed against one of the regulatory subunits, one of the alpha subunits and PA28 a or b. In the context of the study also blots with beta subunit Abs should be included.

lane 59/60.

There is absolutely no experimental evidence that IP degrade proteins in an ATP independent way in vivo. See recent review by Kors et al. *Front Mol Biosci.* 2019 Jul 16;6:48

Reviewer #2 (Comments to the Authors (Required)):

So far it was expected that approved proteasome inhibitors, such as bortezomib, are effective in the treatment of triple-negative breast cancer (TNBC). Adwal et al. demonstrate that expression of the immunoproteasome subunit PSMB8 is proportional to the density of tumor-infiltrating lymphocytes (TILs), whereas it is suppressed in brain metastases. The authors conclude that

inhibiting proteasome activity might be counter-productive in early TNBC treatment.

Comments:

- 1) Summary (p2): The summary is confusing.
- 2) Introduction (p2): The introduction to the UPS system in the first paragraph is not comprehensible for readers who are not familiar with the subject. There are only two references which seem to be chosen incorrectly.
- 3) Intro (p2/3): The individual paragraphs in the introduction should match better.
- 4) Res&Disc (p3): The first paragraph shows how MDA-MB-468 differs from MCG7 in terms of PSMB8 sensitivity. This paragraph is well written, but experiments need to be validated by an expert in these techniques.
- 5) Res&Disc (p3): In the second paragraph several proteins involved in UPR are listed, but no results. Thus, this section should be transferred to the intro.
- 6) Res&Disc (p4): It seems to me that the authors describe their results only in a few sentences, but have a detailed discussion in the following. Even when combining Results and Discussion in the same section, there should be more results than discussions.
- 7) Res&Disc (p5): I am not familiar with IHC analysis. Although, the results seem to be representative 'Of 415 evaluable cases, staining was homogeneously strong in 64.8%, negative in 8.9% and heterogeneous in 26.3%', the assessment by an expert is necessary.
- 8) Res&Disc (p6): 'PSMB8 IHC analysis of 34 brain metastases and patient-matched primary breast tumor samples' have been analyzed and compared to experiments in mice, this looks like to be an elaborate study. Therefore, the conclusion that 'links between defective antigen processing, treatment resistance and metastatic progression may also occur in non-TNBCs' might be justified. However, an expert has to be consulted for verification.
- 9) Conc. (p6): Adwal et al. conclude that 'Inhibiting proteasome activity could be counter-productive in the early TNBC treatment setting, because antigen processing fidelity is linked to the effectiveness of radiotherapy and chemotherapy'. Bortezomib is a non-specific inhibitor. Why didn't the authors test carfilzomib or a specific inhibitor for the immunoproteasome? These additional experiments are strongly recommended.

Reviewer #3 (Comments to the Authors (Required)):

In this work, the authors found that bortezomib induce toxicity in breast cancer cell lines expressing high PSMB8/9/10, immune-proteasome subunit. Further analyses in human TCGA cohorts reveal that the expression of subunit for immuno-proteasome associate with better survival in patient with CT/XRT and higher T cell infiltration. The analyses on cell lines are pretty compelling. However, the human relevance is less clear. The following issues should be addressed to strengthen the conclusion.

1. CT usually compromise T cell-mediated anti-tumor responses when high doses are used. Therefore, this does not support the claim provided by the authors. The possibility on whether immuno-proteasome expressing tumor cells can elicit stronger UPR responses that can induce cell death or senescence upon CT should be considered.
2. Whether the increase of immune-proteasome subunit indeed associate with better antigen presentation should be examined. Moreover, whether patients with high expression of immuno-proteasome subunit have higher mutations (it can lead to higher amount of neoantigens) should be examined in TCGA cohort.

Trade-off between metabolic immuno-proteasome addiction and immune evasion in triple-negative breast cancer

By Alaknanda Adwal, Priyakshi Kalita-de Croft, Reshma Shakya, Emarene Kalaw, Lucinda Taege, Amy McCart Reed, Sunil Lakhani, David Callen and Jodi Saunus.

Dear Dr Leibfried,

We have spent the last few months performing additional experiments to address the reviewers' comments on our manuscript. Please see pages 2-4 for a point-by-point response, and the revised manuscript which has been edited accordingly, including the addition of a substantial amount of new data that supports our previous conclusions. New blocks of text are colored grey.

Your synthesis of this feedback was that there were three major points requiring attention, which I have separately responded to here:

1. *The dataset remains somewhat limited and an extension and further cohort analyses are requested (#1, #3)*
We performed IHC studies on two additional biomarkers (the MHC-I complex and PA28 i-proteasome cap) on the long-term follow-up cohort as well as new genomic analyses looking at potential links with mutation burden.
2. *...testing more specific immunoproteasome inhibitors (#2).*
We performed additional in vitro experiments with Carfilzomib and now include new results that concur with bortezomib experiments.
3. *Reviewer #3 thinks that the conclusion that high expression of the immunoproteasome subunit PSMB8 correlates with survival due to better T-cell infiltration is potentially wrong as chemotherapy usually compromises T cell mediated anti-tumour responses.*

We appreciate this comment and have changed the wording in the manuscript to improve clarity. Yes, breast cancer patients most certainly experience systemic immunosuppression **whilst** undergoing chemotherapy, but the presence of TILs prior to treatment predicts ability to mount effective anti-tumour immune responses once treatment is complete, and treatment-associated immune suppression is lifted. There is now international consensus that the presence and density of TILs in pre-treatment breast tumour biopsies predicts the response to neo-adjuvant chemotherapy, and also the response to second-line treatment in patients with a low-moderate disease burden evident in post-treatment mastectomy tissue.

Thank you for considering our revised manuscript. We have made a concerted effort to strengthen our paper on the basis of this feedback and hope it is now considered worthy of publication in LSA.

Sincerely,

The conclusions of the study are interesting from a clinical point of view the results are in general terms neither surprising nor entirely new. It is known that IP are protective against proteotoxic stress and it is also known that high IP levels enhance surface MHC class I expression which is a prerequisite for improved T cell stimulation.

Our studies produced novel and very clinically-relevant findings that we feel have been overlooked by reviewer-1. The findings are important because, despite the notion that IP activity *should* be associated with more effective antigen presentation, this is still not considered in the rationale for new proteasome inhibitor trials in TNBC (e.g. the AGMT MBC-10 trial, currently underway) or indeed other solid tumour types where there is an immunologic component to treatment response. There is a persisting assumption that this therapy only affects constitutive subunits in solid tumours, as we have argued in the manuscript, highlighting a recent proteasome inhibitor therapy review. ***Our findings have the potential to shape future clinical research activity and investment.***

The 'textbook narrative' asserts that the i-proteasome is induced by IFN γ in lymphocytes and myeloid cells, but we found that breast tumour cells express substantial levels, that its induction by IFN γ has little effect on sensitivity to bortezomib, but rather, that its activity is a consequence of metabolic dependence sustained by copy-number alterations. Hence, IP subunit over-expression in TNBC is driven by genomic events that are positively selected during tumour development. ***These are novel findings relating to the development of a substantial proportion of breast tumours.***

We also integrated findings from multiple independent genomic datasets, and complemented these analyses with IHC studies on another independent breast cancer cohort with more than 25 years' follow-up data, and specifically in brain-metastatic breast cancers with matching brain metastases. Collectively, this is a very powerful and unique clinical sample resource-base that links our findings with the active and exciting new sub-field of breast cancer TILs research.

This feedback highlighted to us that some important implications may be overlooked by the reader, hence we have accordingly expanded the text in particular sections to make these points clearer. We were able to do this quite easily without excessively lengthening the article, since the manuscript was originally prepared as a concise communication for JEM and then transferred to LSA.

... IP function in the context of proteotoxic stress is connected [with] PA28 alpha/beta (PSME1/2) and the formation of hybrid IP (19S-20S-PA28) and 20S-PA28 alpha/beta complexes... whose formation is induced by IFN γ ... Also ERAP1/2 has to be considered in the context to antigen processing.... The expression of PA28 in the context of inhibitors resistance and the analysis of both PA28ab und ERAP expression in TNBC ought to be included in the study as well to complete the picture. A reanalysis of the clinical data sets I would regard as sufficient.

This is difficult question to address in by 'reanalysis of clinical datasets', because there is a strong correlation between tumour cell purity and expression of i-proteasome RNA. Our most conclusive *in silico* PSMB8/9 analyses were based on genomic copy-number alterations, which are specific to the tumour compartment.

The alternative was IHC analysis, which enables visual discrimination between tumour and stromal components. Performing additional IHC studies is a significant undertaking. This reviewer suggested two additional markers (PA28/PSME2 and ERAP1/2). Reviewer 3 requested additional evidence that PSMB8 is associated with antigen presentation (e.g. parallel analysis of MHC-I levels). To address all comments within the limits of our resources, we performed additional IHC studies with two of the three markers suggested (PA28/PSME2 and MHC-I). The results were consistent with existing conclusions and significantly strengthened the manuscript (see changes to Figure-4).

Fig1D: The figure lacks the expression analysis of standard subunits and upon extension of the study the analysis of PA28ab expression.

The aim of this experiment was to compare immunoproteasome subunit expression in representative basal and luminal cell lines. We selected one of the 20 or so other proteasome/activator complex subunits as a comparator, but the main aim of the experiment would have been addressed even without it. We do not think that the addition of extra western data would add value at this point.

Fig1E: ...by which criteria [did] the authors assign the identity of the individual proteasome complexes. What is needed is a lane with purified 30S/26S complexes and 20S complexes and an immunoblot with antibodies directed against one of the regulatory subunits, one of the alpha subunits and PA28 a or b. In the context of the study also blots with beta subunit Abs should be included.

The native in-gel proteasome activity assay is standard and gives a reproducible pattern each time – many examples can be found in the literature. The aim of this experiment was to determine if total proteasome activity is similarly inhibited in cell lines with different levels of i-proteasome expression, which it does.

line 59/60. There is absolutely no experimental evidence that IP degrade proteins in an ATP independent way in vivo.

This part of the introduction has now been replaced with more detailed background on proteasome subunits to support new analysis in Figure-5.

Reviewer #2

1) Summary (p2): The summary is confusing.

2) Intro (p2): The intro to the UPS system is not comprehensible for unfamiliar readers. There are only two references which seem to be chosen incorrectly.

3) Intro (p2/3): paragraphs in the introduction should match better.

We have made some changes to the wording that hopefully improve readability. Since this manuscript was originally prepared as a concise communication to JEM (strict word limit), and redirected to LSA, we were able to expand the text to achieve this.

4) Res&Disc (p3): In the second paragraph several proteins involved in UPR are listed, but no results. This should be transferred to the intro.

We considered this comment carefully, but decided that inclusion of marker details would over-complicate the rationale for the study if included in the introduction.

5) Res&Disc (p4): It seems to me that the authors describe their results only in a few sentences, but have a detailed discussion in the following. Even when combing Results and Discussion in the same section, there should be more results.

We have now expanded the results and discussion text accordingly. The condensed nature of the submitted manuscript was due to length restrictions in JEM's 'brief definitive report' article category.

6) Conc. (p6): Adwal et al. conclude that 'Inhibiting proteasome activity could be counter-productive in the early TNBC treatment setting, because antigen processing fidelity is linked to the effectiveness of radiotherapy and chemotherapy'. Bortezomib is a non-specific inhibitor. Why didn't the authors test carfilzomib or a specific inhibitor for the immunoproteasome? These additional experiments are strongly recommended.

Carfilzomib has the same substrate specificity as bortezomib – b5c and b5i subunits. The IC50 margin for inhibiting b5c and b5i is wider for carfilzomib than bortezomib, but still within the same order of magnitude. We performed additional experiments with carfilzomib, which showed a similar relationship between sensitivity and i-proteasome subunit expression to bortezomib (Fig-1C).

Reviewer #3

1. CT usually compromises T cell-mediated anti-tumour responses when high doses are used. Therefore, this does not support the claim provided by the authors. The possibility on whether immuno-proteasome expressing tumour cells can elicit stronger UPR responses that can induce cell death or senescence upon CT should be considered.

We appreciate the logic in this comment and think the previous wording did not adequately explain our point. While patients actively undergoing chemotherapy experience systemic immunosuppression (hence are vulnerable to opportunistic infections and related complications), the presence of TILs prior to treatment indicates an ability to mount immune responses once this treatment-associated suppression is lifted. There is now international consensus that the presence and density of TILs in diagnostic biopsies predicts the response to neo-adjuvant chemotherapy, and also the response to second-line treatment in patients with a low-moderate disease burden evident in post-treatment mastectomy tissue. We have now amended/expanded the related text in the introduction and elsewhere to clarify.

2. Whether the increase of immune-proteasome subunit indeed associate with better antigen presentation should be examined. Moreover, whether patients with high expression of immuno-proteasome subunit have higher mutations (it can lead to higher amount of neoantigens) should be examined in TCGA cohort.

Our paper already demonstrates that b5i (PSMB8) correlates with multiple markers of anti-tumour immunity, which depends on effective antigen presentation. We showed that b5i correlates with the densities of stromal antigen presenting cells and TILs expressing feedback markers. We have now also added additional evidence (Fig-5B/C): (1) b5i correlates with levels of MHC-I (based on IHC analysis with an antibody that recognises HLA-A/B/C; findings consistent with Lee et al., *Cancer Res Treatment* (2019), which is already referenced in the manuscript); (2) b5i correlates with levels of PA28b (*PSME2*, co-induced by IFN γ but located outside the gained HLA locus); (3) b5i correlates with levels of tumour cell PD-L1, which reflects the degree of TILs engagement (Sobral-Leite et al., *OncImmunology*, 2018).

We followed the reviewer's suggestion and explored a possible link between mutation burden and b5i. This is difficult question to address in the way suggested, because there is a strong correlation between tumour cell purity and PSMB8 RNA expression. We instead used *PSMB8* copy-number gain as a surrogate marker of tumour cell over-expression, and found no significant difference with copy-number-neutral cases in terms of overall burden of copy-number alterations, single nucleotide variants or indels in TCGA datasets. These data are now included in Figure-S4.

April 22, 2020

RE: Life Science Alliance Manuscript #LSA-2019-00562-TR

Dr. Jodi M. Saunus
The University of Queensland
UQ Centre for Clinical Research
B71/918 The Royal Brisbane & Women's Hospital
Herston, Queensland 4029
Australia

Dear Dr. Saunus,

Thank you for submitting your revised manuscript entitled "Tradeoff between metabolic i-proteasome addiction & immune evasion in triple-negative breast cancer". As you will see, the reviewers appreciate the introduced changes, and we would thus be happy to publish your paper in Life Science Alliance pending final minor revisions:

- Please address the remaining reviewer concerns by careful discussion
- Please add callouts in the manuscript text to Fig 5F, S1A,B
- Where missing, please add p-value descriptions in the figure legends next to the statistical test mentioned (eg, Figs 1-4); where missing, please add statistical tests employed in the figure legends
- Please revise your figures - font size is often too small and text not readable at normal figure size; please pay particular attention to Fig S2
- Figure 5 legend misses description of panel G, Fig. S1 legend misses description of panel C, please add
- Please add scale bars to Fig. 1F, 5A, S4A,C,E
- Please provide source data for Fig 1D
- Insets should be added to Fig. 5A and S4, matching the magnifications shown
- Please add a statement in the M&M section on written consent provided by the patients of the two cohorts used

A. FINAL FILES:

B. MANUSCRIPT ORGANIZATION AND FORMATTING:

Sincerely,

Andrea Leibfried, PhD
Executive Editor

Life Science Alliance
Meyerhofstr. 1
69117 Heidelberg, Germany
t +49 6221 8891 502
e a.leibfried@life-science-alliance.org
www.life-science-alliance.org

Reviewer #1 (Comments to the Authors (Required)):

The essential take home message of the significantly improved revised version of the manuscript is that treating tumors (breast tumors) with proteasome inhibitors may be counter productive because it impairs anti-TNBC immunity.

Neglecting somewhat the still relatively poor biochemistry, I can follow the argumentation of the authors based on the bioinformatic data shown. It makes absolutely sense to me that a clinical setting with i-proteasome high/TIL high promises to be of advantage. However, because the conclusions drawn may have

considerable impact on the future treatment of patients I believe that it is important to take one potential artifact into careful consideration which resides in the available data bases.

The validity of the potentially important correlation between i-proteasome high in tumor cells / TIL high in a given tumor depends at large on the purity and reliability the tumor samples analyzed.

With other words, were the breast tumor samples

really devoid of (contaminating) TILs which contain predominantly

immunoproteasomes. Thus, in the worst case high TIL numbers will also give high i-proteasome numbers in tumor cells because of of insufficient separation. I am not making the point that this is the case, but because of the potential importance of the study and being an experimental scientist who is aware of the underlying technical problems in separating tumor tissues I feel that it is in my responsibility to point at potential pitfalls which may easily be overlooked.

Having made this statement I also would like to point out that the revised manuscript contains a number of interesting new data which are of general interest.

Reviewer #2 (Comments to the Authors (Required)):

The authors have satisfactorily addressed my concerns.

Reviewer #3 (Comments to the Authors (Required)):

This revised manuscript addresses my concerns and the revised manuscript is largely improved. It should be published since the context and the findings can be interesting in the relevant fields.

RE: #LSA-2019-00562-T

Trade-off between metabolic immuno-proteasome addiction and immune evasion in TNBC

By Alaknanda Adwal, Priyakshi Kalita-de Croft, Reshma Shakya, Emarene Kalaw, Lucinda Taege, Amy McCart Reed, Sunil Lakhani, David Callen and Jodi Saunus.

Dear Dr Leibfried,

Thank you for confirming acceptance of our article. We are pleased to provide a final manuscript draft that address the minor revisions required:

- Callouts in the manuscript text now added for the following:
 - Fig-S1. The references to parts A, B and C were in the wrong order, apologies. This has now been rectified (highlighted on lines 108 and 118).
 - Fig-5F was also highlighted in the editorial review but we can confirm there are already several references to this in the paragraph beginning on line 225.
- Statistical test details including p-value descriptions have now been added to all figure legend parts.
- Figure font size reviewed and increased throughout (incl. Fig S2)
- Figure 5G and S1C legend descriptions added.
- Scale information added to Fig-1F (50um bars), 5A (legend, line 496), S4 (legend, line 531).
- Source data provided for Fig 1D (raw blots_Fig1D.pdf)
- Insets added to Fig-5A and S4.

Regarding ethical use of the two cohorts used, written consent is not a requirement under either of our human research ethics approvals because these are retrospectively sourced, surplus samples from the archives of our state-wide pathology service. Most samples are from the late 1980s/early 1990s, which provides up to 30 years of clinical follow-up necessary to capture clinical outcomes across the diverse breast cancer population. The ethical rationale for using these samples and associated clinical data without consent is that: (a) the FFPE blocks used represent excess of diagnostic requirements, where tissue was collected as part of standard clinical practice; (b) patients are de-identified to anyone not named on the ethics approval; (c) the results have no bearing on clinical management, and the majority of patients are now deceased; (d) The potential benefits to future patients far outweigh any risk of a privacy breach. We have published many molecular pathology studies under this approval (see footnote).

Finally, reviewer-1 is spot-on about measurements of immunoproteasome subunit RNA expression in tumor tissue homogenates being affected by the presence of TILs. In fact, we openly discuss this being a limitation of tissue homogenate analysis (lines 190-198) and explained how we circumvented this by obtaining readouts that are specific to the tumor cell compartment: gene copy-number aberrations (Fig-4) and in situ analysis of protein (Fig-5). Hence I think this issue is already thoroughly covered, but please let us know if the editorial team thinks there is something else we should do.

Thank you again for considering and agreeing to publish our work.

Footnote. Examples of studies published using the two cohorts of Adwal et al. under the same HREC approval since 2005:

- | | |
|--|---|
| ▪ Al-Ejeh, F., et al., Oncogenesis , 2014. 3: p. e124. | ▪ McCart Reed, A.E., et al., NPJ Breast Cancer , 2019. 5: p. 18. |
| ▪ Field, S., et al., Int J Cancer , 2016. 138(8): p. 1959-70. | ▪ Saunus, J.M., et al., J Pathol , 2015. 237(3): p. 363-78. |
| ▪ Hernandez-Perez, S., et al., Oncogene , 2017. | ▪ Wiegman, A.P., et al., JCI Insight , 2019. 5. |
| ▪ Junankar, S., et al., Nat Commun , 2015. 6: p. 6548. | ▪ Zhang, L., et al., Nature , 2015. 527(7576): p. 100-4. |
| ▪ McCart Reed, A.E., et al., J Pathol , 2019. 247(2): p. 214-227. | |
| ▪ Zhang, L., et al., Science Translational Medicine , 2020. In press . | |

May 12, 2020

RE: Life Science Alliance Manuscript #LSA-2019-00562-TRR

Dr. Jodi M. Saunus
The University of Queensland
UQ Centre for Clinical Research
B71/918 The Royal Brisbane & Women's Hospital
Herston, Queensland 4029
Australia

Dear Dr. Saunus,

Thank you for submitting your Resource entitled "Tradeoff between metabolic i-proteasome addiction & immune evasion in triple-negative breast cancer". I appreciate the introduced changes and it is a pleasure to let you know that your manuscript is now accepted for publication in Life Science Alliance. Congratulations on this interesting work.

DISTRIBUTION OF MATERIALS:

Again, congratulations on a very nice paper. I hope you found the review process to be constructive and are pleased with how the manuscript was handled editorially. We look forward to future exciting submissions from your lab.

Sincerely,

Andrea Leibfried, PhD
Executive Editor
Life Science Alliance
Meyrhofstr. 1
69117 Heidelberg, Germany
t +49 6221 8891 502
e a.leibfried@life-science-alliance.org
www.life-science-alliance.org